# Photodynamic Therapy Using Talaporfin Sodium for Local Failure after Chemoradiotherapy or Radiotherapy for Esophageal Cancer: A Single Center Experience

**DOI:** 10.3390/jcm9051509

**Published:** 2020-05-17

**Authors:** Natsuki Ishida, Satoshi Osawa, Takahiro Miyazu, Masanao Kaneko, Satoshi Tamura, Shinya Tani, Mihoko Yamade, Moriya Iwaizumi, Yasushi Hamaya, Takahisa Furuta, Ken Sugimoto

**Affiliations:** 1First Department of Medicine, Hamamatsu University School of Medicine, 1-20-1 Handayama, Higashi-ku, Hamamatsu 431-3192, Japan; ma03006@hama-med.ac.jp (N.I.); tmiyazu@hama-med.ac.jp (T.M.); mkaneko@hama-med.ac.jp (M.K.); tamura@hama-med.ac.jp (S.T.); miyamade@hama-med.ac.jp (M.Y.); yhamaya@hama-med.ac.jp (Y.H.); sugimken@hama-med.ac.jp (K.S.); 2Department of Endoscopic and Photodynamic Medicine, Hamamatsu University School of Medicine, 1-20-1 Handayama, Higashi-ku, Hamamatsu 431-3192, Japan; s-tani@hama-med.ac.jp; 3Department of Laboratory Medicine, Hamamatsu University School of Medicine, 1-20-1 Handayama, Higashi-ku, Hamamatsu 431-3192, Japan; iwaizumi@hama-med.ac.jp; 4Center for Clinical Research, Hamamatsu University School of Medicine, 1-20-1 Handayama, Higashi-ku, Hamamatsu 431-3192, Japan; furuta@hama-med.ac.jp

**Keywords:** photodynamic therapy, talaporfin sodium, esophageal cancer, chemoradiotherapy, radiation therapy

## Abstract

A phase II study of second-generation photodynamic therapy (PDT) using talaporfin sodium has shown excellent treatment results for esophageal cancer with local failure after chemoradiotherapy (CRT) or radiotherapy (RT). However, only a few studies have reported this therapy in clinical practice. This study aimed to confirm the efficacy and safety of salvage PDT using talaporfin sodium for esophageal cancer in various clinical situations. Twelve patients with esophageal cancer with local failure after definitive CRT or RT who underwent PDT using talaporfin sodium were enrolled from April 2016 to January 2020. Overall, 10 patients (83.3%) achieved a local complete response. No skin phototoxicity was observed, but esophageal stricture occurred in five patients (41.7%). Esophageal stricture was improved with endoscopic balloon dilation in all patients, and subsequent analysis found no significant factors causing esophageal stricture after PDT. Two patients with synchronous tumors were successfully rescued by combination therapy with endoscopic submucosal dissection. Two patients with carcinoma in situ of larger than 1/2 circumference were rescued by repeated PDT. The 2-year overall survival was 80.0% (95% confidence interval 0.409–0.946). PDT using talaporfin sodium was an effective and safe salvage treatment for esophageal cancer with local failure after CRT or RT in various clinical situations.

## 1. Introduction

Definitive chemoradiotherapy (CRT) is a non-surgical curative option in patients with inoperable, locally advanced lesions or in those who decline to be surgically treated even in the operable stage of esophageal cancer [1,2,3,4]. However, a considerable number of patients develop locoregional failure after definitive CRT or RT [1,5], and salvage therapy is necessary for local recurrence or residual lesions [6,7]. Although salvage esophagectomy is often performed with a curative intent, it is associated with high rates of complications and surgery-related mortality [8,9,10]. Photodynamic therapy (PDT) was developed as a cancer treatment that induced tissue destruction by combination with a photosensitizer. PDT is expected to be a more powerful salvage therapy compared with endoscopic resection (ER) after CRT or RT because it is indicated not only for superficial cancer with submucosal fibrosis, but also for tumors in the muscularis propria in cases of no metastasis [11,12,13,14].

First-generation PDT for esophageal cancer using porfimer sodium (Photofirin; Pfizer Japan Inc., Tokyo, Japan) has some disadvantages such as a long sun shade period of approximately 6 weeks to avoid the risk of skin phototoxicity and the need for an expensive excimer dye laser system [15,16,17]. Conversely, second-generation PDT using talaporfin sodium (Laserphyrin; Meiji Seika Pharma Co., Ltd., Tokyo, Japan) requires a shorter sun shade period of approximately 2 weeks and has a low frequency of skin phototoxicity [11,18,19]. Recently, Yano et al. reported the favorable results of a phase II study examining the local complete response (L-CR) of PDT using talaporfin sodium for esophageal cancer with local failure after CRT or RT [20]; according to the results of this clinical study, PDT using talaporfin sodium was approved in Japan at October 2015 [21]. However, only few clinical studies have reported this treatment because it is currently only being performed at limited centers.

To increase the use of this salvage PDT as an established therapy, it is necessary to accumulate practical experiences in various conditions. Therefore, the present study investigated the efficacy and safety of salvage PDT using talaporfin sodium for locally recurrent or residual esophageal cancer after CRT or RT in various clinical practices as well as the treatment of comorbidities and complicated lesions.

## 2. Experimental Section

### 2.1. Patients

A total of 12 patients with local failure after CRT or RT for esophageal cancer who were treated with PDT using talaporfin sodium at Hamamatsu University Hospital from April 2016 to January 2020 were enrolled. The inclusion criteria were as follows: (1) esophageal cancer with local recurrence or residual tumor after CRT or RT; (2) ER, such as endoscopic mucosal resection and endoscopic submucosal dissection (ESD), and surgical treatment was impossible or not expected; (3) invasion depth of T1-2 (within the shallow muscularis propria layer); (4) no invasion to the cervical esophagus; and (5) no apparent lymph node metastasis or distant metastasis. In this study, the restriction of circumference was not considered in the inclusion criteria. Moreover, lesions greater than semi-circumferential lesions were included and treated under the strategy of a planned split of PDT after obtained informed consent in terms of stricture. The clinical stage (cStage) of cancer of the enrolled patients was determined using the tumor–node–metastasis (TNM) classification of the International Union Against Cancer [22]. The clinical T stage was evaluated by endoscopy, computed tomography (CT), and endoscopic ultrasound (EUS), whereas the clinical N and M stages were evaluated using EUS and CT.

The study protocol was reviewed and approved by the ethics committee of Hamamatsu University School of Medicine (no. 18–165: 26 September 2018). The study was conducted in accordance with the Declaration of Helsinki. Written informed consent was obtained from all patients prior to participation in PDT.

### 2.2. PDT Procedure

The PDT procedure was performed according to a previous report [20]. The diode laser (PD-LASER, PHC Holdings Corporation, Japan) was delivered via a frontal light distributor through the operative channel of an endoscope (EG-590WR2, FUJIFILM Holdings Corporation, Japan). The plastic cap was attached to the tip of endoscope to maintain the distance between the scope and the lesion. Laser irradiation was performed at 664 nm and 100 J/cm^2^ for 4–6 h after intravenous administration of a 40 mg/m^2^ dose of talaporfin sodium. The next day, endoscopic observation was performed and patients were treated with additional irradiation if necessary. All the patients were instructed to stay in a room maintained at ≤500 lux for 2 weeks after PDT. A skin photosensitivity test was performed 2 weeks after the administration of talaporfin sodium, and the patients were discharged if the skin photosensitivity and other adverse events related to PDT disappeared. Endoscopic examination was performed 1 week after PDT, and further endoscopy was performed at the discretion of each attending physician.

### 2.3. Outcome Measures

The local efficacy of PDT was assessed based on the local complete response (L-CR), which was the primary endpoint. L-CR was defined as follows: (1) absence of an obvious residual tumor; (2) disappearance of post-PDT ulcers; and (3) histological confirmation of the absence of cancer cells by biopsy. Biopsy was not performed in all patients if no obvious residual lesion was observed. If additional PDT was performed repeatedly to the residual lesion, L-CR was judged after the last PDT. The secondary endpoints were adverse events related to PDT using talaporfin sodium, overall survival (OS), and progression-free survival (PFS). OS and PFS were defined as the period from the date of PDT to death and progression, respectively.

### 2.4. Statistics

Statistical analysis was performed using the SPSS (SPSS for Windows, Version 16.0; SPSS Inc., Chicago, IL, USA) and EZR (Saitama Medical Center, Jichi Medical University, Saitama, Japan) software. Categorical data were analyzed using χ2 test and Fisher’s exact test. Data for survival outcomes were calculated using the Kaplan–Meier method. *p* < 0.05 was considered to be statistically significant.

## 3. Results

### 3.1. Patient Characteristics

The baseline characteristics of the patients at initial CRT or RT are shown in Table 1. Among the 12 patients, seven (58.3%) were men and five (41.7%) were women. All the patients were diagnosed with squamous cell carcinoma (SCC), and eight (66.7%) had cStage I at initial treatment. Overall, eight (66.7%) patients were treated by CRT and the remaining four patients by RT.

The patient characteristics at salvage PDT are shown in Table 2. Overall, 10 (83.3%) patients had recurrent lesions and the remaining two had residual lesions. As shown in Table 2, there were three patients with T1a (mucosal cancer) at PDT. It was difficult to treat these lesions by ER due to scars after CRT or RT for initial esophageal cancer, which was confirmed by EUS. In one patient, the main lesion (20 cm from the incisors) involved a submucosal-invasive cancer in part and circumferential SCC in situ. Because there was no other treatment option, he was treated with a planned split of PDT following informed consent.

### 3.2. Efficacy and Survival

The median total irradiation dose was 400 J, and the median hospital stay was 19 days (Table 3). The median follow-up duration in this study was 14 months. Overall, 10 patients (83.3%) achieved L-CR, and the 2-year OS after PDT was 80.0% (95% confidence interval [CI], 0.409–0.946) (Figure 1). One patient died due to mediastinal lymph node metastasis from the primary esophageal cancer. However, recurrent or residual lesions were not observed at the primary lesion treated by PDT. Another patient died from another disease after being transferred to another hospital.

The PFS after PDT was 72.7% (95% CI, 0.371–0.903). Residual lesions were still observed in two patients, and the local efficacy was not complete response. In one patient, a residual lesion with deep ulceration was diagnosed by forceps biopsy 4 months after PDT; this patient was unable to be treated with additional PDT due to deep ulceration, and he developed an esophagobronchial fistula 5 months after PDT, associated with the residual lesion. In another patient, a residual lesion was detected by forceps biopsy 2 months after PDT, which could be rescued with an additional salvage PDT.

### 3.3. Safety

The safety of PDT using talaporfin sodium in terms of adverse events is shown in Table 4. No skin phototoxicity was observed in the patients in this study. Esophageal strictures due to PDT occurred in five patients, all of whom required endoscopic balloon dilation (EBD) for the strictures. Most of the patients had esophageal strictures with dysphagia at approximately 4 months (range: 1–9) and required multiple EBDs (range: 2–10) for improvement in passage obstruction (Appendix A). As described above, esophagobronchial fistula due to residual lesion was observed in one patient, and because this patient could not continue oral intake to avoid aspiration pneumonia, he received tube feeding by percutaneous endoscopic gastrostomy (PEG) 7 months after PDT.

We investigated the factors associated with esophageal stricture after PDT. As shown in Table 5, no significant factors were noted between the stricture group and the non-stricture group, although a circumferential ulcer 1 month after PDT (*p* = 0.06) and CRT at initial treatment (*p* = 0.08) tended to be a factor for stricture.

### 3.4. Combination Therapy of Repeated PDTs and ESDs

In this study, the type of local failure after CRT or RT varied in number, size, depth, and location. Given that diversity is one of the features of post-CRT and post-RT recurrent tumors, we further evaluated whether PDT was able to overcome such diversity. As shown in Table 6, two patients had large lesions and were able to achieve L-CR as a result of repeated PDTs. In addition, two patients with multiple recurrent tumors were successfully treated by combining PDT and ESD.

One of the above patients required both repeated PDT and ESDs (Figure 2); this patient had a circumferential residual cancer after CRT in the upper esophagus (20 cm from the incisors) and two other superficial cancers in the middle esophagus (25 and 30 cm from the incisors). Given that it was considered difficult to treat all the lesions at once, only the central part of the lesion was irradiated in the first PDT under the strategy of a planned split of PDT (Figure 2a,b,f). L-CR was achieved after three rounds of PDT. Esophageal stricture due to PDT occurred as an adverse effect and was improved by EBDs (Figure 2c,i). The other two lesions were treated with two rounds of ESDs (Figure 2d,e,g,h).

## 4. Discussion

The present study aimed to confirm the efficacy and safety of the novel PDT using talaporfin sodium, a second-generation photosensitizer, for esophageal cancer with local failure after definitive CRT or RT in various conditions. According to our results, the L-CR rate was 83.3% and 2-year OS was 80.0%. No skin phototoxicity was observed, whereas esophageal stricture occurred in five patients (41.7%) that could be improved by EBD. We also confirmed the usefulness of combination therapy with ESD for multiple lesions and repeated PDT therapy for widespread lesions. These findings support the results of the previous phase II study, thereby indicating the usefulness and widespread use of salvage PDT for esophageal cancer after local failure.

To the best of our knowledge, this is the fourth report of salvage PDT using talaporfin sodium for esophageal cancer [19,20,23]. According to the previous phase II study in Japan [20], salvage PDT with talaporfin sodium showed an excellent high L-CR rate (88.5%; 95% CI, 69.8%–97.6%; T1, 100% [21/21]; T2, 57.1% [4/7]). A previous study reported no cases of treatment-related mortality and no cases of skin phototoxicity within the 2-week sun shade period [20]. In another subsequent study, Minamide, et al. reported that salvage PDT using talaporfin sodium showed a better L-CR rate than PDT using porfimer sodium (69.0% vs. 58.1%) and the common complications of skin phototoxicity, esophageal stricture, and esophageal fistula were less frequent in the talaporfin sodium group than in the porfimer sodium group [23]. Although the number of patients in the present study was small, the L-CR rate was reasonable compared with that reported in the previous reports. The absence of skin phototoxicity observed in the present study was also consistent with that reported in previous studies. Although salvage esophagectomy is a curative treatment option, it is associated with a high rate of complications and surgery-related mortality [8,9,10]. In cases of local failure without metastases, we believe that the safety and effectiveness of PDT would be superior to those of salvage surgery, although there are no data regarding the direct comparison of these treatment options.

In our study, esophageal stricture occurred in five patients (41.7%) after salvage PDT; this number is slightly higher than that reported in previous studies. We tried to determine the risk factors for esophageal stricture due to PDT. According to a comparative analysis between two groups of patients with and without esophageal stricture, we found no significant risk factors between the two groups. However, a circumferential ulcer at 1 week after PDT tended to be a factor for stricture, which appeared to be a reasonable result. Furthermore, in our small cohort, all the patients who developed esophageal stricture were treated with CRT at initial treatment, whereas in the group of patients without esophageal stricture, three patients were treated with CRT and the remaining three patients were treated with RT. Although the number of patients was too small to make any definitive conclusions, CRT as an initial treatment also tended to be a risk factor for developing esophageal stricture after PDT.

EBD was an effective treatment option for esophageal stricture after salvage PDT in our patients. EBD reportedly improved esophageal stricture after surgery with a single treatment in approximately 20% of patients, and multiple EBDs were required to improve the stricture [24,25]. Recently, intralesional steroid injection or systemic steroid therapy has been reported to prevent stricture after ESD for esophageal cancer [26,27,28]. However, to our knowledge, the effects of steroid therapy for the prevention of esophageal stricture after PDT have not been studied to date. Therefore, we have performed EBDs without steroid therapy in our patients with post-PDT stricture. Given that frequent EBD increases patient distress, the development of a better method to avoid such distress is expected. Recently, the fitting of black plastic attachment is recommended to avoid esophageal stricture by a reduction in the diffused reflection of diode laser.

In this study, esophagobronchial fistula was observed as a serious complication after PDT in one patient; this patient had residual cancer with deep ulceration and additional salvage treatment, including repeated PDT, was impossible. An esophagobronchial fistula was noted in this patient at 5 months after PDT associated with a residual lesion; thus, the patient could no longer receive oral administration and PEG was required. In this patient, it was difficult to determine whether the esophagobronchial fistula was an adverse event associated with PDT or the result of cancer progression.

In our case series, the type of local failure after CRT or RT varied in both number and size. There were two patients who had multiple recurrence in the esophagus and three patients who had lesions larger than 1/2 circumference or ≥3 cm. Although diversity is one of the features of post-CRT and RT failure, salvage PDT using talaporfin sodium combined with ESD was useful for multiple lesions, and widespread lesions could be treated by repeated PDT. These findings suggest that salvage PDT using talaporfin sodium is a flexible option that could overcome a wide variety of disease states. Furthermore, the shorter shading period was acceptable, even for elderly patients aged >80 years.

There are several limitations in this study. First, the number of patients studied was small because only a limited number of patients are eligible for this therapy and this study was performed as a preliminary study in a single-center setting. Second, the outcome was investigated by short-term evaluation as was not long enough, since this therapy became available in clinical practice, and whether salvage PDT for esophageal cancer contributes to long-term prognosis remains unclear. Further accumulation of cases and long-term follow-up details of the clinical course of this therapy are expected to clarify these issues. Therefore, multicenter prospective controlled trials are required in the future.

Recognizing these limitations, the implications of our report are that this treatment was able to reproduce the results of the phase II trial in various real-world settings without difficulty and that this therapy can be used flexibly even in elderly patients. To date, the aging of population has been progressing and elderly patients with esophageal cancer are more likely to be treated with CRT or RT alone, instead of surgery, for comorbidities and treatment risks [29,30]. Salvage PDT is reported to be a safer and more effective treatment option than other minimally invasive treatments such as re-irradiation or endoluminal brachytherapy in elderly patients [31,32]. Finally, we hope that salvage PDT using talaporfin sodium will be recognized as a promising salvage treatment for esophageal cancer with local failure after CRT and function as a treatment of choice in the future.

## Figures and Tables

**Figure 1 jcm-09-01509-f001:**
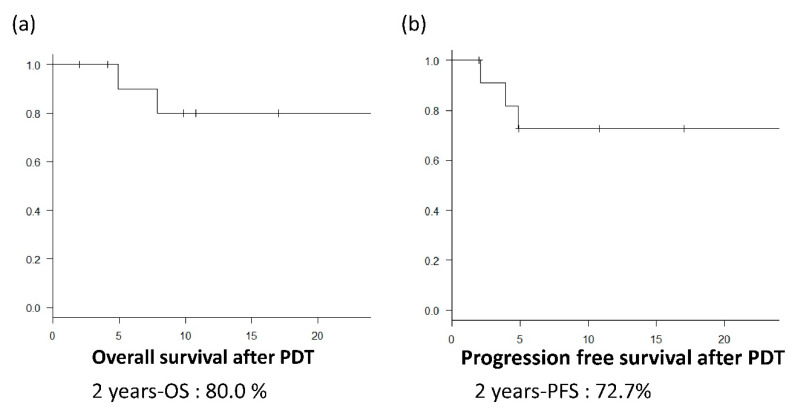
Overall survival (OS) and progression-free survival (PFS). (**a**) Overall survival (OS) curve after photodynamic therapy (PDT) using talaporfin sodium; the 2-year OS was 80.0% (95% confidence interval (CI), 0.409–0.946). (**b**) PFS after PDT using talaporfin sodium; the 2-year PFS was 72.7% (95% CI, 0.371–0.903).

**Figure 2 jcm-09-01509-f002:**
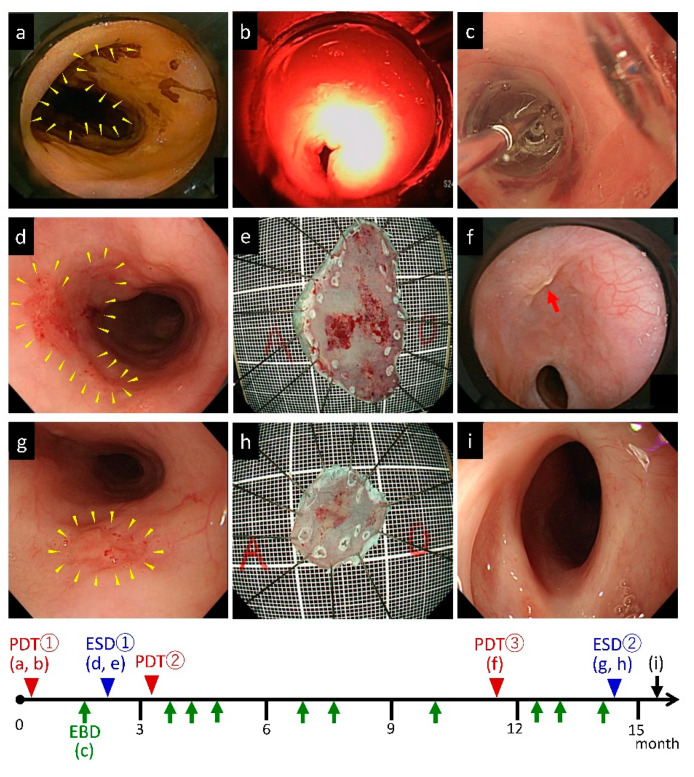
A patient required both repeated photodynamic therapy (PDT) for widespread lesion and endoscopic submucosal dissection (ESD) for the other two lesions. This patient had a circumferential residual esophageal cancer after CRT in the upper esophagus (20 cm from the incisors) and two other superficial cancers in the middle esophagus (25 and 30 cm from the incisors). (**a**) Main lesion (20 cm from the incisors) involving a submucosal-invasive cancer in part and circumferential squamous cell carcinoma (SCC) in situ (arrowhead). (**b**) The first PDT was performed to the main lesion. Because it was considered difficult to treat all lesions at once, only the central part of the lesion was irradiated in the first PDT under the strategy of a planned split of PDT. (**c**) The esophageal stricture after PDT was treated with endoscopic balloon dilation (EBD). (**d**) A second lesion of approximately 2 cm in size was observed in the middle esophagus (25 cm from the incisors, arrowhead). (**e**) The second lesion was treated by ESD. (**f**) The main lesion after second PDT. The residual small lesion—SCC in situ—was treated with repeated PDT (red arrow). (**g**) A third lesion of approximately 1 cm in size was observed in the middle esophagus (30 cm from the incisors, arrowhead). (**h**) The third lesion was treated by ESD. (**i**) L-CR was achieved after three rounds of PDT for the main lesion and ESDs for the other two lesions. The esophageal stricture after PDT was improved by EBDs.

**Table 1 jcm-09-01509-t001:** Patient characteristics at prior treatment.

Number of Patients		*n* = 12
Sex (male:female), *n* (%)		7:5 (58.3:41.7)
Age (mean ± SD, range, year) at PDT		74.7 ± 8.3 (63–87)
Tumor location	upper, *n* (%)	4 (33.3)
	middle, *n* (%)	6 (50.0)
	lower*, n* (%)	2 (16.7)
Histological type		SCC
cStage at prior treatment	cStage I, *n* (%)	8 (66.7)
	cStage II, *n* (%)	2 (16.7)
	cStage III, *n* (%)	1 (8.3)
	cStageIV_A_, *n* (%)	1 (8.3)
Prior treatment (CRT: RT), *n* (%)		8:4 (66.7:33.3)
Regimen of chemotherapy	CDDP + 5-FU	7
	CDGP + 5-FU	1
Total dose of radiotherapy (median, range, Gy)		60 (50.4–70)

SD: standard deviation, SCC: squamous cell carcinoma, cStage: clinical stage, CRT: chemoradiotherapy, RT: radiotherapy, CDDP: cisplatin, CDGP: nedaplatin, PDP: photodynamic therapy.

**Table 2 jcm-09-01509-t002:** Patient characteristics at PDT.

		*n* = 12
Tumor status	recurrent tumor	10
	residual tumor	2
Number of lesions	single	10
	multiple	2
Invasion depth at PDT	T1a	3
	T1b	9
Circumference of lesion at PDT	≤1/4	6
	>1/4, ≤1/2	4
	>1/2, ≤3/4	1
	>3/4, ≤1	1
Longitudinal lesion length (mean ± SD, range, cm)		1.67 ± 0.86 (1–4)
Interval between CRT/RT and PDT (median, range, months)		9 (3–192)

PDT: Photodynamic therapy, SD: Standard deviation, CRT: Chemoradiotherapy, RT: Radiotherapy.

**Table 3 jcm-09-01509-t003:** Treatment outcomes of PDT.

		*n* = 12
Total dose of irradiation (median, range, J)		400 (200–800)
Total number of PDT (median, range, times)		1 (1–3)
Hospital stay (median, range, days)		19 (17–28)
Local efficacy	L-CR, *n* (%)	10 (83.3)
	L-nonCR, *n* (%)	2 (16.7)

PDT: photodynamic therapy, L-CR: local complete response, L-nonCR: local non-complete response.

**Table 4 jcm-09-01509-t004:** Adverse events related to PDT.

Adverse Event	*n* = 12	Treatment
Skin phototoxicity, *n* (%)	0 (0.0)	
Esophageal stricture, *n* (%)	5 (41.7)	All patients needed EBD
Esophagobronchial fistula, *n* (%)	1 (8.3)	PEG 7 months after PDT

PDT: photodynamic therapy; EBD: endoscopic balloon dilation; PEG: percutaneous endoscopic gastrostomy.

**Table 5 jcm-09-01509-t005:** Comparison between the esophageal stricture and non-stricture groups.

	Stricture *n* = 5	Non-Stricture *n* = 7	*p*-Value
Age (mean ± SD, range, years)	73.8 ± 6.22	75.3 ± 9.98	0.78
Sex (male:female)	3:2	5:2	0.56
Invasion depth at prior treatment (T1:T2, 3, 4)	3:2	6:1	0.52
Prior treatment (CRT:RT)	5:0	4:3	0.08
Radiation dose (mean ± SD, range, Gy)	62.0 ± 4.47 (60–70)	61.5 ± 6.79 (50.4–70)	0.89
Invasion depth at PDT (T1:T2, 3, 4)	5:0	7:0	
Circumference of lesion (mean ± SD, range)	0.42 ± 0.21 (1/4–1)	0.35 ± 0.11 (1/4–1/2)	0.46
Longitudinal lesion length (mean ± SD, range, cm)	2.00 ± 1.22 (1–4)	1.43 ± 0.45 (1–2)	0.28
Total irradiation dose per round (mean ± SD, range, J)	475 ± 196 (227–700)	457 ± 181 (200–800)	0.87
Black attachment use, n (%)	1 (20.0)	4 (57.1)	0.22
Alb at PDT (mean ± SD, range, g/dL)	4.16 ± 0.05 (4.1–4.2)	3.74 ± 0.60 (2.8–4.4)	0.16
Hb at PDT (mean ± SD, range, g/dL)	11.5 ± 1.02 (10.1–12.6)	11.9 ± 1.81 (9.6–14.4)	0.64
CRP at PDT (mean ± SD, range, mg/dL)	0.07 ± 0.03 (0.03–0.12)	1.33 ± 2.35 (0.03–6.57)	0.27
Circumferential ulcer 1 week after PDT (mean ± SD, range)	1.00 ± 0 (1)	0.75 ± 0.29 (1/4–3/4)	0.06

CRT: chemoradiotherapy, RT: radiotherapy, PDT: photodynamic therapy, Alb: albumin, Hb: hemoglobin, CRP: C-reactive protein.

**Table 6 jcm-09-01509-t006:** Details of patients treated with repeated PDT.

Patient	Age, Year	Sex	PDT	Tumor Location	Circumference of Lesion at PDT	Tumor Depth	Total Dose of Irradiation, J	Outcomes	Adverse Effects
1	69	M	1st	Upper	1	SM	400	---	Stricture
			2nd	Upper	1/3	M	300	---	Stricture
			3rd	Upper	≤1/4	M	400	L-CR	Stricture
2	65	F	1st	Lower	2/3	SM	500	---	Non
			2nd	Lower	1/3	SM	700	L-CR	Non

PDT: photodynamic therapy, L-CR: local complete response.

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
