# Peer review of "Photodynamic Therapy Using Talaporfin Sodium for Local Failure after Chemoradiotherapy or Radiotherapy for Esophageal Cancer: A Single Center Experience"

_jcm, 2020, doi:10.3390/jcm9051509_

Round 1

Reviewer 1 Report

This report described a clinical feature of 12 patients who received PDT after failure for CRT or RT in the clinical practice.

  1. The indication of PDT in this manuscript is unclear. Please describe more details about depth and circumference.
  2. Six patients with T1a underwent PDT. You should describe why ER was not indicated for these patients.
  3. In table 2, the baseline and outcome of the patients were mixed. Please describe these in separate tables.
  4. Please clarify the median follow up time of this analysis.
  5. I agree with your conclusion on the usefulness of salvage PDT for the patients. In clinical practice, salvage surgery is one of the modality with curative intent. Please discuss the safety and efficacy of PDT comparison with salvage surgery.

Author Response

Response to Reviewer 1

  1. The indication of PDT in this manuscript is unclear. Please describe more details about depth and circumference.

Author’s response”

We are very grateful to Review 1 for these critical comments and thoughtful suggestions. We hope that the revised manuscript along with our responses can now be accepted for publication.

Based on your comment, we have added details about depth and circumference in the manuscript: “invasion depth of T1-2 (within the shallow muscularis propria layer)” in the Experimental Section, 2.1. Patients (page 2, line 71) and “In this study, the restriction of circumference was not considered in the inclusion criteria. Moreover, lesions greater than semi-circumferential lesions were included and treated under the strategy of a planned split of PDT after obtained informed consent in terms of stricture.” in the Experimental Section, 2.1. Patients (page 2, lines 72–75).

  1. Six patients with T1a underwent PDT. You should describe why ER was not indicated for these patients.

Author’s response

There were 3 patients with T1a (mucosal cancer) at PDT, and this information has been included in Table 2. It was difficult to treat these lesions by ER due to scars after CRT or RT for initial esophageal cancer, which was confirmed by EUS. Accordingly, we have added the following sentences in revised manuscript (page 3, lines 119–121) “As shown in Table 2, there were 3 patients with T1a (mucosal cancer) at PDT. It was difficult to treat these lesions by ER due to scars after CRT or RT for initial esophageal cancer, which was confirmed by EUS.”

  1. In table 2, the baseline and outcome of the patients were mixed. Please describe these in separate tables.

Author’s response

Based on your comment, we have separated Table 2 into Patient characteristics at PDT (Table 2) and patient outcomes (Table 3).

  1. Please clarify the median follow up time of this analysis.

Author’s response

The median follow-up duration in the present study was 14 months. We have added the follow-up duration in the outcome measure section (page 4, line 133).

  1. I agree with your conclusion on the usefulness of salvage PDT for the patients. In clinical practice, salvage surgery is one of the modalities with curative intent. Please discuss the safety and efficacy of PDT comparison with salvage surgery.

Author’s response

Thank you for your positive feedback. According to your comment, we have added the description about the safety and efficacy of PDT vs. salvage surgery in the Discussion section “Although salvage esophagectomy is a curative treatment option, it is associated with a high rate of complications and surgery-related mortality [8-10]. In cases of local failure without metastases, we believe that the safety and effectiveness of PDT would be superior to those of salvage surgery, although there is no data regarding the direct comparison of these treatment options.” (page 8, lines 225–229).

Reviewer 2 Report

In this manuscript, Ishida et al. report 12 cases of locally failed esophageal cancer patients who underwent PDT using talaporfin sodium. They found that PDT using talaporfin sodium was an effective and safe salvage treatment. Their report is important for this new therapy to become a candidate for the treatment of locally failed esophageal cancer after CRT/RT.

Minor points:

  1. L-CR in the Abstract needs to be spelled out.
  2. Figure 1 in the Figure legend is in italic.
  3. “5. Conclusions” in the end of Discussion should be deleted.
  4. Acknowledgements section can be deleted.
  5. Minor grammatical errors are seen in the text.

Author Response

Response to Reviewer 2

  1. L-CR in the Abstract needs to be spelled out.

Author’s response

We are very grateful to Review 2 for these critical comments and thoughtful suggestions. We hope that the revised manuscript along with our responses can now be accepted for publication.

According to your comment, we have defined the abbreviation L-CR (local complete response) in the abstract (page 1, line 25).

  1. Figure 1 in the Figure legend is in italic.

Author’s response

We apologize for any overlooks and have accordingly revised the font.

  1. “5. Conclusions” in the end of Discussion should be deleted.

Author’s response

We have deleted “5. Conclusions” in the end of Discussion in the revised manuscript.

  1. Acknowledgements section can be deleted.

Author’s response

Based on your suggestion, we have deleted the Acknowledgements section in the revised manuscript.

  1. Minor grammatical errors are seen in the text.

Author’s response

The manuscript has been checked by a native English proofreader.

Reviewer 3 Report

this is an interesting and relevant study dealing with non surgical options for the management of local recurrence following radiation therapy for Squamous Cell cancers of the oesophagus.

some reference to other options including re-irradiation and or intra luminal brachytherapy in this context would be strengthen the manuscript and its relevance in this cohort of patients. the authors are correct to highlight the need for these options to exist in light of the aging population and morbid surgical salvage which limits its utility

there needs to be some improvement in the grammar and presentation of the manuscript and I would recommend a proof read from an English speaker would be beneficial.

Author Response

Response to Reviewer 3

  1. Some reference to other options including re-irradiation and or intra luminal brachytherapy in this context would be strengthen the manuscript and its relevance in this cohort of patients. the authors are correct to highlight the need for these options to exist in light of the aging population and morbid surgical salvage which limits its utility

Author’s response

We are very grateful to Review 3 for these critical comments and thoughtful suggestions. We hope that the revised manuscript along with our responses can now be accepted for publication.

Although intraluminal brachytherapy and re-irradiation performed in the past are minimally invasive, their therapeutic effects are less than that of PDT. As the reviewer has commented, comparison with these treatments will further emphasize that PDT is a minimally invasive and useful treatment for the elderly. We have added the following sentences in the Discussion section “Salvage PDT is reported to be a safer and more effective treatment option than other minimally invasive treatments such as re-irradiation or endoluminal brachytherapy in elderly patients.” (page 9, lines 277–279).

  1. There needs to be some improvement in the grammar and presentation of the manuscript, and I would recommend a proof read from an English speaker would be beneficial.

Author’s response

The manuscript has been checked by a native English proofreader.